# Lola-I is a promoter pioneer factor that establishes de novo Pol II pausing during development

Vivekanandan Ramalingam [1,2,5], Xinyang Yu[3], Brian D. Slaughter [1], Jay R. Unruh [1], Kaelan J. Brennan[1], Anastasiia Onyshchenko[1], Jeffrey J. Lange[1], Malini Natarajan[1], Michael Buck [3,4] & Julia Zeitlinger [1,2] ✉

While the accessibility of enhancers is dynamically regulated during development, promoters tend to be constitutively accessible and poised for activation by paused Pol II. By studying Lola-I, a *Drosophila* zinc finger transcription factor, we show here that the promoter state can also be subject to developmental regulation independently of gene activation. Lola-I is ubiquitously expressed at the end of embryogenesis and causes its target promoters to become accessible and acquire paused Pol II throughout the embryo. This promoter transition is required but not sufficient for tissue-specific target gene activation. Lola-I mediates this function by depleting promoter nucleosomes, similar to the action of pioneer factors at enhancers. These results uncover a level of regulation for promoters that is normally found at enhancers and reveal a mechanism for the de novo establishment of paused Pol II at promoters.

Gene regulation during development depends on the coordinated action of enhancer and promoter sequences. Enhancers respond to specific developmental signals and transmit the information to the core promoter where transcription of the gene begins[1]. Since promoters must respond to a large variety of different enhancers, they tend to have constitutively accessible chromatin[2–4]. In contrast, distally located developmental enhancers typically only become accessible in the cell lineage in which they become active[1,5]. To prevent activation in inappropriate cell types, developmental enhancers tend to be wrapped around stable nucleosomes in their natural state and thus are inaccessible to most transcription factors[6–8]. This nucleosome barrier is most often overcome with the help of so-called pioneer factors that recognize their DNA-binding motifs on nucleosomal DNA and initiate chromatin remodeling[9]. Once enhancers are accessible, other transcription factors can bind and drive enhancers toward activation.

Promoters on the other hand use a variety of mechanisms to maintain their accessibility across cell types and conditions. Some inherently have a low nucleosome barrier due to nucleosome-disfavoring sequences (e.g., poly-A tracts)[10–12], while others are actively kept accessible through constitutively expressed pioneer factors, which antagonize the main nucleosome over the promoter. Examples of such constitutively expressed promoter pioneer factors are Reb1 or Abf1 in budding yeast, GAGA factor (GAF) in *Drosophila*, and SP1 in mammals[13–17].

A hallmark of accessible promoters is the presence of paused RNA polymerase II (Pol II), which is present even in cell types where the genes are inactive[18–24]. At such promoters, Pol II initiates transcription and transcribes for 30–50 bp before going into a paused state, and the presence of paused Pol II poises genes for robust induction during development[18,19,25,26]. Upon induction, paused Pol II is released into productive elongation and new Pol II initiates at high frequencies, which results in a burst of transcription[27–29].

While the majority of promoters are constitutively accessible and have paused Pol II[3,4,19–22], promoters could nevertheless be regulated, by acquiring paused Pol II only in a specific developmental context or

[1]Stowers Institute for Medical Research, Kansas City, MO, USA. [2]Department of Pathology and Laboratory Medicine, University of Kansas Medical Center, Kansas City, KS, USA. [3]Department of Biochemistry, State University of New York at Buffalo, Buffalo, NY, USA. [4]Department of Biomedical Informatics, Jacobs School of Medicine & Biomedical Sciences, Buffalo, NY, USA. [5]Present address: Department of Genetics, Stanford University, Palo Alto, CA, USA. ✉e-mail: jbz@stowers.org

at alternative start sites. Indeed, how alternative start sites are regulated is generally not known[30]. The mechanisms by which promoters might undergo such a transition and how this might affect the target gene expression has not been studied.

By analyzing promoters with high levels of paused Pol II in *Drosophila*, we previously observed that a subset of promoters had no Pol II occupancy during the beginning of embryogenesis (2–4 h after egg laying - AEL, referred as "early"), but acquired paused Pol II gradually over time until the end of embryogenesis (14–17 h AEL, referred as "late")[19]. Since *Drosophila* is a well accessible experimental system, this provided us with an opportunity to discover the mechanisms by which promoters acquire Pol II de novo during development. Here, we identify one of the responsible transcription factors as Lola-I and found that Lola-I functions at promoters analogous to pioneer factors at enhancers. By binding to its DNA-binding motif near the nucleosome edge, it makes its target promoters broadly accessible throughout the embryo, without directly causing gene activation. In this manner, Lola-I orchestrates the de novo acquisition of paused Pol II over the course of embryogenesis, producing poised promoters ready for tissue-specific transcriptional bursts during differentiation. These results illustrate a new mechanism for the regulation of promoters, thus providing a basis for studying promoter regulation in mammals.

## Results

### Lola-I is required for paused Pol II and chromatin accessibility at target promoters over developmental time

To discover how paused Pol II is acquired de novo at the late stages of embryogenesis (14–17 h), we considered that the process is regulated by a transcription factor and searched for DNA-binding motifs that are enriched at these promoters (opening set)[19]. As a control, we used promoters with high levels of paused Pol II throughout embryogenesis (constant set)[19]. The most highly enriched motif was AAAGCT (>6-fold enrichment; Supplementary Data 1) (Fig. 1a), a motif bound by the zinc-finger transcription factor Lola-I[31].

Lola-I is encoded by one of the more than 25 different splice isoforms from the *lola* locus[32]. All Lola proteins code for transcription factors that share the same N-terminal BTB/POZ domain, but have isoform-specific C-terminal zinc-finger domains with distinct DNA-binding specificities and developmental roles[32–35]. Notably, the RNA of Lola-I is upregulated during the later stages of embryogenesis[36], consistent with our proposed role for Lola-I.

To test whether Lola-I indeed specifically binds to the promoters in the opening set, we raised polyclonal antibodies specific for the Lola-I isoform (Supplementary Fig. 2a). These antibodies confirmed that the Lola-I protein strongly increased in abundance during the late stages of embryogenesis (Fig. 1b). We then used these antibodies to perform chromatin immunoprecipitation experiments coupled to sequencing (ChIP-seq) on late *Drosophila* embryos (14–17 h). Among the top 750 regions, 725 had a Lola-I motif within 200 bp of the peak summits (MEME motif enrichment E-value < 1e−561), confirming that the antibodies were specific for Lola-I.

Around 60% of the Lola-I-bound regions mapped to annotated promoters (Supplementary Data 2), and these promoters were those that acquired paused Pol II over time and showed increased chromatin accessibility in DNAse-seq data (Fig. 1c, d, and Supplementary Fig. 1a). In comparison, control promoters from the constant set[19] did not show this increase (Fig. 1d, Supplementary Fig. 1a). Finally, we found that the Lola-I motifs in these promoters often showed specific conservation across *Drosophila* species, which is indicative of their functional importance (Supplementary Fig. 1a, b).

To test whether Lola-I is the factor responsible for causing these promoters to change over developmental time, we analyzed a previously identified *lola-I* mutant, *lola*[ORC4][32]. This mutant contains a premature stop codon specifically in the *lola-I* isoform, predicting a truncated Lola-I protein without the zinc-finger DNA-binding domain

(Fig. 1e). A low amount of truncated Lola-I version was indeed detected in *lola*[ORC4/ORC4] embryos by Western blot, while immunostainings showed essentially a complete loss of signal (Fig. 1e, and Supplementary Figs. 2a, 4c). These results confirm the identity of the *lola-I* mutants and the specificity of the Lola-I antibodies. By balancing the mutant chromosome over a GFP-marked balancer, we were able to isolate large numbers of *lola-I*[−/−] embryos through an automated large-particle sorter (COPAS).

When we performed Pol II ChIP-seq experiments on *lola-I*[−/−] embryos, we found that Pol II occupancy was specifically reduced at Lola-I targets (median mut/wt signal = 0.44), but not at control promoters (Fig. 1f, g). Furthermore, Lola-I targets showed reduced chromatin accessibility as measured by ATAC-seq (median mut/wt signal = 0.46) (Fig. 1g), and promoters with the strongest reduction in ATAC-seq signal were also those with the strongest reduction in Pol II occupancy (Supplementary Fig. 6c). Both Pol II pausing and chromatin accessibility were not completely abolished, suggesting that other factors, perhaps other Lola isoforms, may help increase the accessibility of the promoters in the absence of Lola-I (Supplementary Fig. 6c). However, the observed changes were not due to a developmental delay and were specific to Lola-I. They were not due to secondary mutations in the *lola*[ORC4] line since they were also observed when heterozygous over another *lola-I* null allele, *lola*[ORES0] (Supplementary Fig. 2b). Importantly, the loss of accessibility and paused Pol II was rescued by the transgenic expression of Lola-I (Fig. 1f, g). This demonstrates that Lola-I directly mediates changes at its target promoters, which leads to the acquisition of paused Pol II and chromatin accessibility.

### Lola-I establishes paused Pol II throughout the embryo but mediates tissue-specific gene expression

The simplest explanation for the observed effect of Lola-I on Pol II is that Lola-I is a strong activator that opens promoters and leads to increased levels of Pol II recruitment and productive elongation. If this were the case, binding of Lola-I would be expected to correlate both temporally and spatially with the expression of its target genes. On the other hand, occupancy of Lola-I and paused Pol II in tissues where the target genes are not expressed would argue that Lola-I establishes paused Pol II independently of gene activation.

To distinguish between the two scenarios, we first analyzed where Lola-I is expressed in the embryo. In immunostainings, nuclear Lola-I was found ubiquitously throughout the embryo (Fig. 2a, Supplementary Fig. 4a). Using single-molecule RNA fluorescence in situ hybridization (single-molecule FISH), *lola-I* RNA was also detected ubiquitously in the embryos (Supplementary Fig. 4b). With Lola-I being ubiquitous, we next asked whether paused Pol II was also acquired ubiquitously across all tissues. To isolate specific tissues from late-stage embryos, we used the INTACT method[18,37,38] to genetically tag the nuclei from the tissue of interest and isolate the nuclei with the help of streptavidin-coupled magnetic beads. We performed Lola-I and Pol II ChIP-seq experiments on isolated nuclei from neurons, muscle, trachea, and epidermis and gut as described previously[18].

Lola-I binding and paused Pol II were present at target promoters across all examined tissues. This was true even for tissue-specific genes (Fig. 2b, c). For example, based on in situ hybridization of whole-mount embryos, the target gene *Gip* is specifically expressed in crystal cells, an immune cell type found near the proventriculus. However, paused Pol II is present at this loci in all tissues, not just in crystal cells (Fig. 2b). This is not due to heterogeneity or low purity of isolated tissues since control genes show the expected tissue specificity of elongating Pol II (*Osi20* in Fig. 2b)[18]. This suggests that Lola-I changes the promoter state without necessarily activating gene expression.

To confirm the ubiquitous effect of Lola-I on promoters and the tissue-specific target gene expression, we analyzed Lola-I ChIP-seq and Pol II ChIP-seq data more globally and compared the data to single-cell

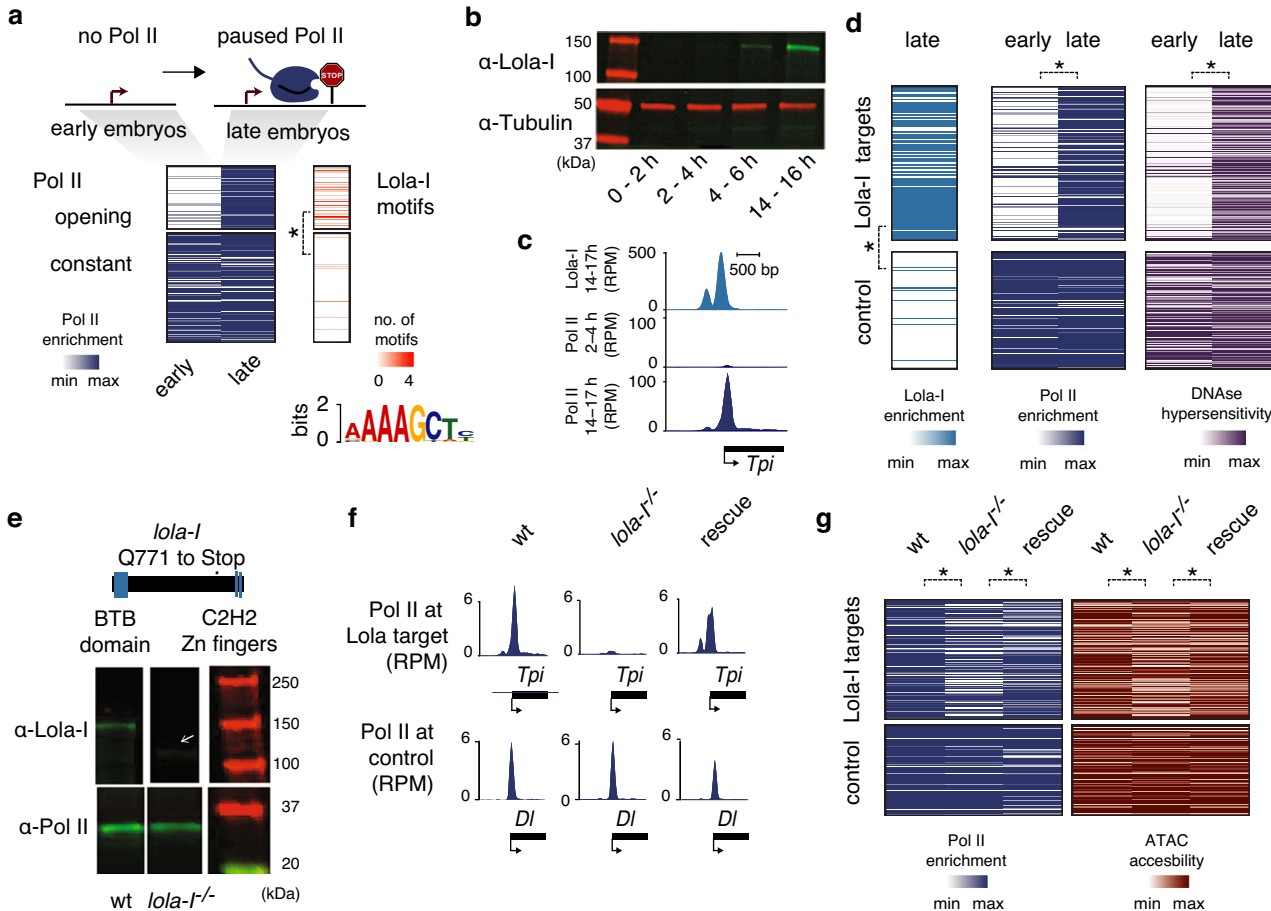

**Fig. 1 | Lola-I is required for the loading of paused Pol II to target genes in the late *Drosophila* embryo. a** At promoters that have no Pol II occupancy in the early (2–4 h AEL) embryo but acquire paused Pol II in the late (14–17 h AEL) embryo (opening set), the Lola-I motif was identified de novo within 200 bp upstream of the TSS by MEME analysis (e-value = 8.1e−091). The Lola-I motif was enriched 6.6-fold in the opening set ($n = 492$) vs the constant set ($n = 843$) (*$P$ = 8.75e−34, chi-squared test with multiple-testing correction). **b** A Western blot with antibodies specific for the Lola-I isoform shows that Lola-I increases in expression during embryogenesis. Tubulin is shown as control. Results shown are representative of at least comparable biological experiments and are consistent. Source data is provided as a Source Data file. **c** Single-gene example of the ChIP-seq data showing that Lola-I binding is found at the promoter of *Tpi*, a gene that acquires paused Pol II over time. **d** Heatmaps showing that the -60% of Lola-I-bound regions found at promoters ($n = 329$) is associated with an increase in Pol II occupancy, RNA levels from the early (2–4 h AEL) to the late (14–17 h AEL) embryo, and DNAse hypersensitivity from the early (stage 5/3 h AEL) to the late (stage 14/11 h AEL) embryo[92]. A random sample of 250 promoters from the constant set[19] is used as control. The

star denotes significance (*$P < 2$e−16) using a two-sided Wilcoxon test. **e** Mutant line *lola-I*[ORC4] [32] has a premature stop codon before the C2H2 zinc-finger region that codes for the DNA-binding domain. A Western blot confirms that *lola-I*[ORC4] homozygous embryos produce a low amount of truncated Lola-I product (white arrow). The Rpb3 subunit of Pol II is shown as control below. The wt and *lola*[−/−] lanes were not run adjacently in the original gel. Results shown are representative of at least two biological replicates and are consistent. Source data is provided as a Source Data file. **f** Pol II ChIP-seq signal at the Lola-I target gene *Tpi* is strongly reduced in homozygous *lola*[ORC4] mutant embryos, while the control gene *Dl* remains unchanged. In the rescue line, which expresses *lola-I* cDNA in the *lola-I*[ORC4] mutant background, Pol II occupancy is rescued. **g** Heatmap showing that the Pol II ChIP-seq signal and ATAC-seq accessibility is specifically reduced at the Lola-I target promoters in *lola-I*[ORC4] mutant embryos. In the rescue line, Pol II occupancy and accessibility are rescued to levels comparable to wild-type. The star denotes significance (Pol II – wt-mutant: *$P$ = 6.0e−15, mutant-rescue: *$P$ = 2.3e−4; ATAC– wt-mutant: *$P$ = 1.6e−10, mutant-rescue: *$P$ = 2.8e−08) using a two-sided Wilcoxon test. RPM: normalized reads per million.

---

RNA-seq data (scRNA-seq) from an equivalent stage. As expected, Lola-I target genes showed Pol II occupancy very broadly among cell types, as is known for paused genes[18]. These promoters also had high chromatin accessibility[18], which was dependent on Lola-I in each of the specific tissues we isolated from *lola-I*[−/−] embryos (Supplementary Fig. 2c). In contrast, the expression of the Lola-I target genes was highly tissue-specific, similar to control genes with tissue-specific Pol II[18] (Fig. 2c). This suggests that Lola-I establishes paused Pol II at target promoters throughout the embryo, and that the tissue-specific expression of the genes is acquired separately, through tissue-specific enhancers that are located in the promoter region or further distally. These results rule out the idea that Lola-I is simply a strong activator.

Since Lola-I binding is not sufficient to induce the expression of target genes, we next asked whether Lola-I is nevertheless required

for their expression. This would imply that Lola-I-induced chromatin accessibility and paused Pol II poise promoters for tissue-specific gene expressions. By performing bulk RNA-seq experiments on *lola-I*[−/−] and wild-type embryos, we found that the expression of Lola-I target genes, but not that of control genes, was significantly reduced in *lola-I*[−/−] embryos (Fig. 3a, left panel). Furthermore, the reduction was more strongly associated with the Lola-I-bound promoters than the genes associated with Lola-I-bound distal regions, confirming Lola-I's main effect on promoters (Fig. 3a, right panel). The affected target genes were enriched for functions in metabolism and ion transport (Supplementary Data 3). Such functions are consistent with the mutant phenotype of *lola-I*[−/−] embryos, which fail to hatch at the end of embryogenesis but lack any visible gross abnormalities in neuronal, muscle and glial structures (Supplementary Fig. 4d).

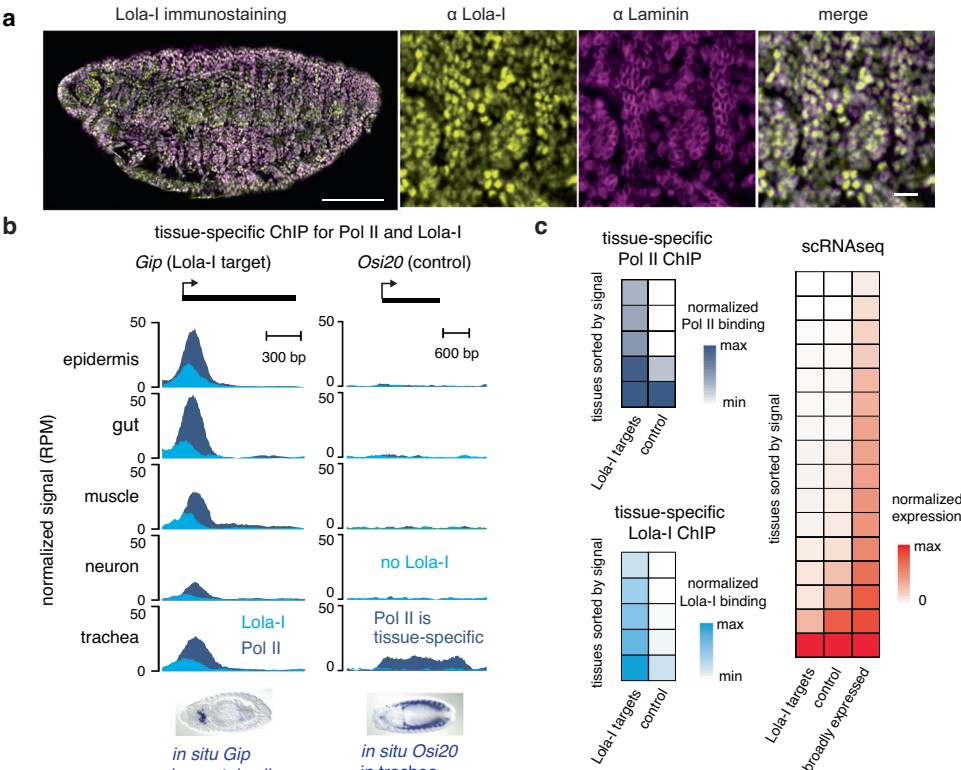

**Fig. 2 | Lola-I establishes paused Pol II throughout the embryo.**
**a** Immunostaining using the Lola-I antibodies (yellow) shows that Lola-I is expressed ubiquitously in the late (14–17 h AEL) *Drosophila* embryo. As control, Lamin is shown (pink), which is also ubiquitous and nuclear. The brightness and contrast settings of the lookup table are linearly adjusted for clarity. The settings in the individual panels are the same as in the merge. scale bar: left - 100 μm and right - 10 μm. **b** ChIP-seq data shown as normalized reads per million (RPM) from isolated embryonic tissues of late-stage embryos (14–17 h AEL) using either Lola-I antibodies (turquoise) or Pol II antibodies (dark blue) reveal that Lola-I binding and paused Pol II are found in all examined tissue at Lola-I targets, even when the target gene is expressed only in a specific tissue (*Gip* is shown as example). Ubiquitous Pol II is not found for all genes, e.g., for the control gene *Osi20*, Pol II binding and expression is restricted to tracheal cells. The tissue-specific expression of *Gip* in crystal cells and *Osi20* in tracheal cells are known from in situ

hybridization shown below, courtesy of Berkeley *Drosophila* Genome Project[93,94].
**c** Average Pol II occupancy, Lola-I occupancy, and scRNA-seq levels across tissues confirm that Lola-I target genes show paused Pol II broadly across tissues but show tissue-specific gene expression, similar to the expression of previously described control genes that have restricted Pol II occupancy across tissues[18]. The enrichment of the Pol II ChIP-seq signal was calculated for each promoter over input, and values of <2 fold were set to 0 (min). For each tissue, the values were then sorted from low to high and normalized to the highest value (max). These sorted values were then averaged across tissues for either the Lola-I targets or the control, showing that the values extend much broader across tissues for the Lola-I targets than for the control. The same procedure was used to depict the scRNA-seq expression values. The Lola-I binding signal was calculated in the same way, except that the values from all genes were normalized to the same maximum signal in order to show that the control genes have lower Lola-I binding.

Since the Lola-I target gene expression was reduced in *lola-I*[−/−] embryos, we wondered whether this loss specifically stems from the tissue where the gene is normally expressed at a high level, or whether the transcript loss could also come from changes in basal expression in other tissues. This is plausible since inactive promoters with paused Pol II typically have detectable basal transcript levels[18,19,39,40]. We performed scRNA-seq experiments, and found that indeed both the tissue-specific expression and the basal levels of target genes were reduced in *lola-I*[−/−] vs wild-type embryos (Fig. 3b, Supplementary Fig. 3a–c). For example, *Gip*'s high expression in crystal cells and its basal expression in other tissues were both reduced in *lola-I*[−/−] mutants (Fig. 3c). Within the limits by which low expression of genes can be confidently compared between two scRNA-seq samples, these results suggest that Lola-I impacts the basal activities of promoters and their ability to produce tissue-specific transcripts.

## Lola-I's effect on promoters increases the gene activation frequency
If Lola-I is not a traditional activator and primarily affects the promoter state, we wondered how Lola-I regulates the ability of promoters to produce tissue-specific transcripts. Lola-I's recruitment of paused Pol II could be a requisite for the activation process, or it could be affecting

how many transcripts are produced upon gene activation. The transcription of most genes occurs in bursts, characterized by a period of transcriptional activity (ON state), during which Pol II produces a burst of multiple nascent RNAs (with a rate constant $K_{prod}$), followed by a period of inactivity (OFF state)[41–44]. Given this two-state model, Lola-I could regulate the rate of gene activation (through the activation rate constant $K_{on}$) and increase the burst frequency. This is typically how enhancers regulate transcription[29,44–47]. Alternatively, Lola-I could regulate the burst size (through the rate constants $K_{prod}$ and $K_{off}$), which can be a promoter-specific property[48–50]. If Lola-I affects the burst size, we would expect that most cells in *lola-I*[−/−] embryos show a similar proportional reduction of Lola-I target gene expression. On the other hand, if Lola-I affects the burst frequency, we would expect a more heterogeneous reduction of transcripts in *lola-I*[−/−] embryos, where some cells in the mutant embryos show high levels of RNA similar to wild-type, while others show very few or no RNA.

To test this, we performed single-molecule FISH[51,52] on the target gene *Gip*, which is bound by Lola-I at the promoter, but not at a nearby distal region. We used a series of fluorophore-labeled small probes against *Gip*, as well as control probes against *PPO1* and *PPO2*, which are not Lola-I target genes but are also specifically expressed in crystal cells. We found that in wild-type late-stage embryos (12–14 h), *Gip* was

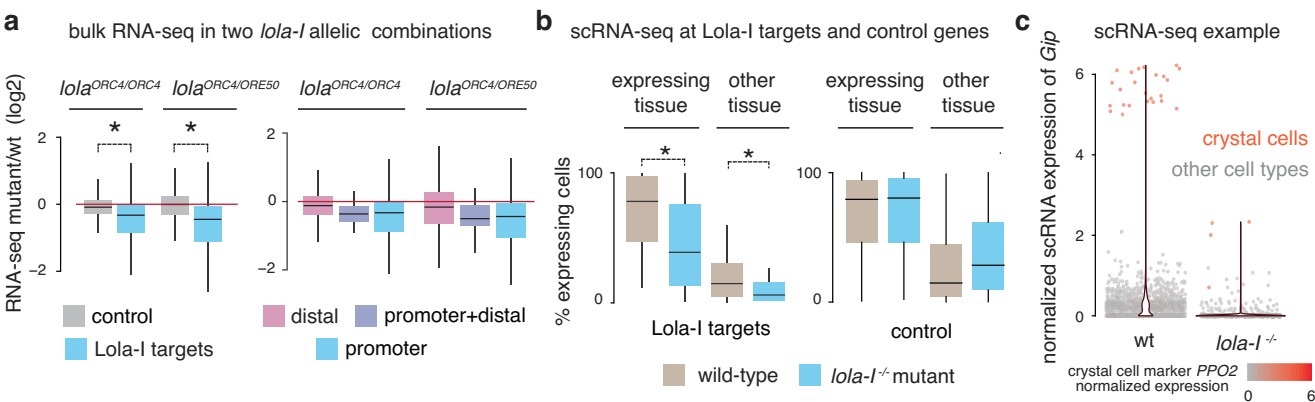

**Fig. 3 | Lola-I promotes tissue-specific gene expression through an effect on the promoter. a** Left panel: Boxplot of bulk RNA-seq data show that Lola-I targets are down-regulated in *lola-I* mutants compared to wild-type (14–17 h AEL), while control genes show overall similar expression levels. This confirms that Lola-I mediates its effect by directly binding to target promoters, and not by indirect effects that would affect all genes (Wilcoxon two-sided test, $lola^{ORC4/ORC4}$: $*P = 1.8e-11$, $lola^{ORC4/ORE50}$: $*P < 2e-16$). Right panel: Boxplot of bulk RNA-seq data show that genes associated with distal Lola-I binding sites are not significantly affected in *lola-I* mutants compared to wild-type (14–17 h AEL), while genes with Lola-I binding at the promoter (with or without distal binding) are reduced in expression, showing that Lola-I largely mediates its effect through binding promoters. **b** scRNA-seq analysis of wild-type and *lola-I* mutant embryos at 14–14.5 h AEL suggests that Lola-I is required for both tissue-specific expression (left) and basal expression levels in other tissues (right) at target genes (defined by >2-fold loss in Pol II occupancy in

mutant embryos). Expression, shown as the fraction of cells with detectable expression, is reduced in *lola-I* mutant embryos compared to wild-type (Wilcoxon two-sided test, expressing tissue: $*P = 1.1e-05$, other tissue: $*P = 0.00578$). The same trend is observed for the median expression (Supplementary Fig. 3c). **c** scRNAseq expression data for the Lola-I target gene *Gip* is shown for each cell (red and gray dots) isolated from wild-type (left) or *lola-I* mutant embryos (right). A violin plot of the data is laid on top. In wild-type embryos, *Gip* is expressed at high levels in crystal cells (marked in red by the expression of the *PPO2* gene), but also shows basal levels of expression in the majority of cells where *Gip* is not expected to be expressed. In *lola-I* embryos, both the specific expression and the basal expression are reduced. Box plots show the median as the central line, the first and the third quartiles as the box, and the upper and lower whiskers extend from the quartile box to the largest/smallest value within 1.5 times of the interquartile range.

expressed at high levels and localized to the same cells as *PPO1/PPO2* (Fig. 4a, b, and Supplementary Fig. 5a), confirming expression in crystal cells. We then estimated the number of *Gip* RNAs for each cell by measuring the cell's total fluorescence intensity and dividing it by the average intensity of individually measured RNAs (example in Fig. 4a inset), which yielded an average of 720 *Gip* RNAs per cell. We also observed clearly detectable bright spots at the sites of nascent transcription (Fig. 4a inset). These spots were present in 41% of the crystal cells and were not observed in other cell types.

In *lola-I*⁻/⁻ embryos of the same stage and analyzed in the same way as wild-type embryos, *Gip* expression was notably reduced, but the reduction was not uniform across all crystal cells. While most *PPO1/PPO2*-positive crystal cells showed little to no detectable *Gip* expression, typically a few cells still showed strong *Gip* expression, albeit lower than wild-type levels (the cells with the top 10% of signal have on average 309 RNAs) (Fig. 4a, b, and Supplementary Fig. 5a). Consistent with this, only 3.5% of the *PPO1/PPO2* positive cells showed bright spots of nascent *Gip* transcription in *lola-I*⁻/⁻ embryos, compared to 41% in wild-type (Fig. 4c). Hence, much fewer mutant crystal cells were actively transcribing *Gip*, but those that were had a substantial number of transcripts.

To rule out that the reduced *Gip* expression was due to developmental delays, we performed the same single-molecule FISH experiments in wild-type and *lola-I*⁻/⁻ embryos over several time points (10–12 h, 12–14 h, 14–16 h). This confirmed the reduced number of cells with *Gip* expression in *lola-I*⁻/⁻ embryos across all time points (Fig. 4b, Supplementary Fig. 5b). Since the reduction in *Gip* expression was highly heterogeneous across cells, the results suggest a reduced burst frequency.

We next fitted our data from wild-type and *lola-I*⁻/⁻ embryos to a simple two-state model[53], using a mathematical framework to fit the parameters $K_{on}$, $K_{off}$ and $K_{prod}$ to steady-state transcript measurements with a fixed RNA degradation rate[52] (Fig. 4d). Since we have measurements for the fraction of cells in the ON state from the nascent transcription spots, we fixed the ratio between the state transition rate

constants ($K_{on}$ and $K_{off}$) to the ratio of cells with and without nascent transcripts. The two-state model produced a reasonably good fit (Fig. 4e, chi squared goodness of fit for wild-type: 4.46 and for mutant: 3.59), and thus did not justify the added complexity of a three-state model[50,54]. We note that this does not eliminate the possibility of a three-state model—it simply indicates that the two-state model is sufficient to adequately fit the data.

The most striking difference between the models from wild-type and *lola-I*⁻/⁻ embryos was the activation rate constant $K_{on}$, which together with $K_{off}$ determines the burst frequency[44]. We observed in the mutants a 13.8-fold reduction of $K_{on}$, which amounts to a reduction of burst frequency of 8.5 fold. Other changes were much smaller (1.4 fold increase in $K_{off}$, 2.7 fold reduction in $K_{prod}$, which amounts to a reduction in burst size of 3.7 fold). We note that the model also suggests a reduction in the burst size, which would be consistent with previous observations that the burst size moderately decreases with lower transcriptional levels[44,55,56]. However, we note that very few cells are producing detectable levels of *Gip* in the mutant, thus it is difficult to accurately assess $K_{prod}$ or burst size (Supplementary Fig. 5d). A live imaging system (such as the MS2-MCP) could provide a better estimate of these values. Whether or not the burst size is moderately affected, our results rule out that Lola-I primarily affects the burst size and show that its strongest effect is on the activation rate and burst frequency. This is remarkable since Lola-I functions in all tissues, and not just in the tissue where the gene is activated. It suggests that Lola-I's effect on the promoter is a requisite for the activation process that enhancers use to produce tissue-specific transcripts.

### Lola-I is a developmentally regulated promoter pioneer factor

To understand the mechanism by which Lola-I changes the promoter state leading to the acquisition of paused Pol II, we considered whether Lola-I is a pioneer factor, which implies that Lola-I removes nucleosomes. By removing the promoter nucleosome, Lola-I would increase chromatin accessibility and allow Pol II recruitment and pausing. Pioneer factors such as GAGA factor have been shown to have such a role

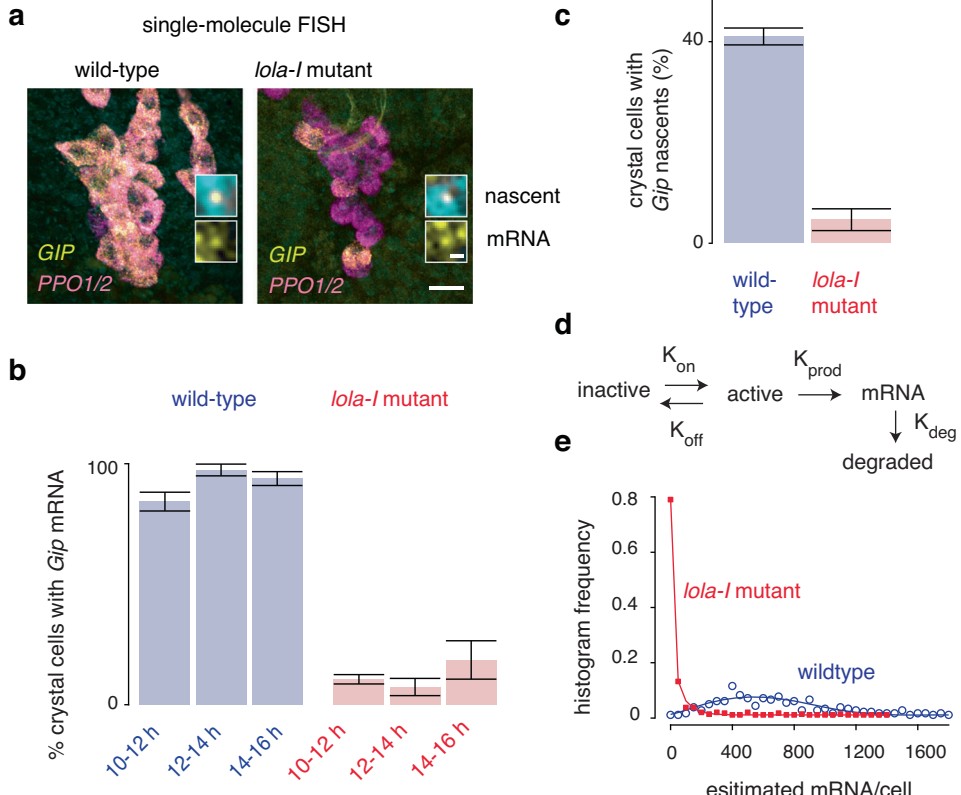

**Fig. 4 | Single-molecule FISH reveals that Lola-I lowers the gene activation barrier. a** Example of single-molecule FISH images of *Gip* (yellow), and *PPO1* and *PPO2* genes (pink) in wild-type embryos and *lola-I*⁻ᐟ⁻ mutant embryos (DAPI - blue). Data were acquired with ×40 magnification, scale bar - 20 μm, with the same brightness and contrast settings. Top insets show the nascent *Gip* signal in the nuclei; bottom insets show the single *Gip* RNAs in the cytoplasm. The brightness and contrast in each inset was adjusted linearly for clarity. **b** The percentage of crystal cells (defined by high expression of *PPO1* and *PPO2*) that express high levels of *Gip* is strongly reduced in *lola-I* mutant embryos compared to wild-type. Barplots show the mean, and the whiskers show the standard error of the mean, with the number of embryos (n): wild-type: 10–12 h *n* = 7, 12–14 h *n* = 8, 14–16 h *n* = 4, *lola-I* mutant: 10–12 h *n* = 7, 12–14 h *n* = 7, 14–16 h *n* = 7. Reduced *Gip* expression is observed for multiple time points showing that it is not due to a developmental delay and that *Gip* expression only mildly recovers over time. **c** The percentage of *PPO1* and *PPO2* positive crystal cells with a visible *Gip* nascent site of transcription (indicative of a transcriptional burst) is also strongly reduced in *lola-I* mutants (12–14 h AEL). The bar represents the mean, and the whiskers show the standard error of the mean (wild-type *n* = 7 and *lola-I* mutant *n* = 14, where n is the number of embryos). **d** Two-state model that was fitted to the data. **e** Histograms of mRNA/cell (dots), calculated from fluorescent intensities at ×100 magnification (see Methods), and fitted lines. The ratio of $K_{off}$ and $K_{on}$ was fixed to the ratio of crystal cells with visible nascent transcripts between wild-type and mutants as shown in (**c**).

at constitutively open promoters[15,16], but such a mechanism has not been described for developmentally regulated promoters. To test this idea, we performed MNase experiments on early and late *Drosophila* embryos in both wild-type and *lola-I* mutant embryos. We found that Lola-I target promoters showed high nucleosome occupancy in the early embryo and a decrease in the late embryo that was Lola-I-dependent (Fig. 5a)(median mut/wt signal = 1.28).

While these results support our hypothesis of Lola-I as a pioneer factor, it is also possible that Lola-I primarily functions to recruit Pol II, and that the nucleosome depletion is a secondary effect of paused Pol II keeping the nucleosome away[23,57]. To distinguish between these two scenarios, we took advantage of the distal Lola-I-bound regions. These regions also contain conserved Lola-I motifs (Supplementary Fig. 1a, b), suggesting that they are functional. If nucleosomes can be removed by Lola-I, they should also be removed in these distal regions without the help of paused Pol II. Indeed, at both promoters and distal regions, Lola-I binding was associated with a strong depletion of nucleosomes (Fig. 5b). Nevertheless, paused Pol II was only detected at promoters, with decreasing Pol II levels the further the distance to the nearest transcription start site (TSS) (Fig. 5c, Supplementary Fig. 6a). This suggests that Lola-I primarily serves to deplete nucleosomes and that the recruitment of Pol II is a secondary step at promoters.

By definition, pioneer factors are able to access their motifs in nucleosomal DNA. They may do so by binding DNA at a particular position on the nucleosome, to linker DNA between nucleosomes, or at the edge of nucleosomes to DNA that becomes accessible when nucleosomes spontaneously unwrap[9,58–61]. We therefore asked whether Lola-I motifs have a preferred position on nucleosomes. Using the MNAse-seq data from early embryos where the promoter nucleosomes are still present, we found a trend of Lola-I motifs to be found at the edge of nucleosomes (Supplementary Fig. 6b). Based on these results, we set out to analyze the binding preference of purified Lola-I protein to nucleosome-bound DNA using a traditional in vitro binding assay combined with high-throughput sequencing[62]. Multiple DNA variants, each with a distinct position of the Lola-I motif embedded in a strong nucleosome-positioning sequence (Widom 601), were reconstituted with nucleosomes in vitro and incubated with different concentrations of full-length Lola-I protein (Fig. 5d, e). The bound and unbound nucleosomal fractions were then sequenced.

The results show that Lola-I bound strongest when the motif was located near the nucleosome edge (R6, R6.5, R7 in Fig. 5d, e and Supplementary Fig. 6e, f) and weakest when the motif was found near the dyad (R0, R0.5, R4, R4.5 in Fig. 5d, e and Supplementary Fig. 6e, f). We did not find strong synergistic effects when multiple Lola-I sites

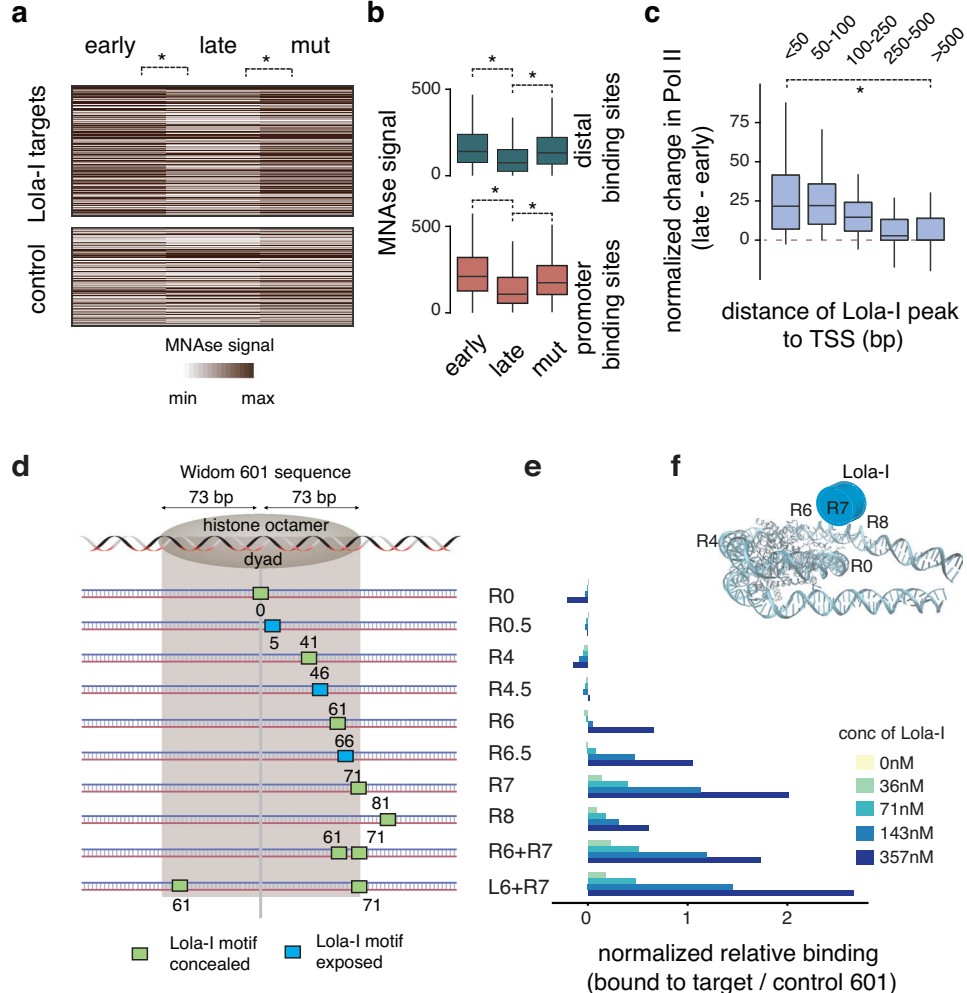

**Fig. 5 | Lola-I is a promoter pioneer factor. a** Heatmap of normalized MNase-seq data shows specific loss of nucleosome occupancy at the Lola-I target promoters during the late stages of *Drosophila* embryogenesis (Wilcoxon two-sided test, *P wt-early vs wt-late: 3.4e−9, wt-late vs mut-late 14–17 h: 3.4e−8). This does not occur in *lola-I* mutant embryos, as there is no statistically significant difference between wild-type early embryos and late *lola-I* mutant embryos. **b** Box plots of the MNase signal (centered on the Lola-I peaks) are shown for promoter-proximal regions and distal Lola-I-bound regions (Supplementary Data 2), which both show a significant decrease in late wild-type embryos (Wilcoxon two-sided test, *P distal, wt-early vs wt-late: *P = 4.5e−09, promoter wt-early vs wt-late: *P < 2e−16, distal wt-late vs mut-late: *P = 1.1e−08, promoter wt-late vs mut-late: *P = 4.1e−10). **c** Pol II recruitment, shown as change in normalized Pol II enrichment from the early to the late stage, only occurs when the Lola-I peak is found in close proximity to the TSS, showing that Lola-I depletes nucleosomes independently of Pol II recruitment (Wilcoxon two-sided test, *P = 1.2e−8). **d** Lola-I binds its motif preferentially at the edge of a

nucleosome in vitro. Purified full-length Lola-I protein expressed in insect cells by baculovirus is incubated in vitro at different concentrations with nucleosomes reconstituted with Lola-I binding sites embedded in the 601 Widom sequence at different positions. **e** The relative binding of the Lola-I protein to the 601-templates containing the Lola-I motif vs the control 601-template without the motif is measured as the loss of signal in the nucleosomal band or the gain of signal in the super-shifted band (Supplementary Fig. 6) (relative to the no Lola-I lane - see methods). The results show that Lola-I binds most strongly when the motif is located along the nucleosomal edge. Significant cooperativity with additional motifs is not observed. **f** Illustration of the preferred position of the Lola-I motif with regard to the nucleosome structure. The nucleosome structure is from the RSCB protein data bank (5NL0). Box plots in the figure show the median (central line), the first and the third quartiles (box), and the largest/smallest value within 1.5 times of the interquartile range as whiskers.

were present (Fig. 5d, e and Supplementary Fig. 6e, f). Moreover, binding to the nucleosome edge occurred whether or not the Lola-I motif was facing outside to the solvent side or was predicted to be concealed inside (Fig. 5d, e and Supplementary Fig. 6e, f), suggesting that Lola-I binds when the DNA is partially unwrapped. Interestingly though, the variant with the Lola-I motif near the accessible linker region (R8) was not bound the highest, indicating that Lola-I does not necessarily prefer free DNA but may also interact with the nucleosome. These results suggest that Lola-I can best access its motif on nucleosomes when located near the edge (Fig. 5f), consistent with our preliminary observations in vivo. Taken together, the results support the conclusion that Lola-I is a developmentally regulated promoter pioneer factor.

## Discussion

In this study, we set out to identify the mechanism by which a set of promoters[19] acquires paused Pol II de novo over the course of embryonic development. Promoters are generally considered to be constitutively open and to have paused Pol II. Hence, we wondered what regulatory step controls the acquisition of Pol II pausing and how this change relates to gene activity. We found that the promoters are targeted by the zinc-finger transcription factor Lola-I ubiquitously throughout the late embryo to establish paused Pol II, but that the target genes are only transcribed in specific tissues.

Our study therefore shows that Pol II transcription can depend on multiple limiting steps. It is often debated whether Pol II initiation or pause release are the limiting regulatory steps for gene expression, and

this is difficult to distinguish as they depend on each other[27,63]. Here we report an altogether different regulatory step that occurs at inactive promoters, before the gene is induced by tissue-specific activation signals. Lola-I functions to transition the promoter from a closed state to a poised promoter state, which allows Pol II to initiate, pause and produce basal levels of expression, but it does not by itself induce a burst of transcription. While Lola-I's effect is not sufficient for gene activation, it is nevertheless required since our single-molecule FISH analysis suggests that Lola-I is specifically required at the level of gene activation. Since the activation rate is controlled by tissue-specific enhancers, these results suggest that Lola-I makes promoters permissive to the tissue-specific activation signals from enhancers.

An obvious explanation for the increased responsiveness of Lola-I target promoters to activation signals is that Lola-I promotes the acquisition of paused Pol II. Paused Pol II is associated with more synchronous and robust gene activation[18,25,26], with higher interaction frequencies with enhancers[64] and localization to the active compartment[65]. Furthermore, the Downstream Promoter Element (DPE) sequence, a hallmark of promoters with Pol II pausing, increases the burst frequency[49,66]. However, Pol II pausing is highly correlated with promoter accessibility, thus we cannot rule out that some of these properties are at least in part due to increased chromatin accessibility. For example, budding yeast does not have Pol II pausing, and yet promoters that have nucleosome-repelling sequences or motifs for pioneer factors have increased promoter strength[67,68]. These results suggest that paused Pol II and promoter accessibility could both increase the rate of gene activation.

The importance of promoter accessibility became apparent when we analyzed the mechanism by which Lola-I establishes paused Pol II. Rather than recruiting Pol II, we found that Lola-I functions at promoters primarily as a pioneer factor that depletes nucleosomes. This is supported by our observation that Lola-I also depletes nucleosomes at distal regions where paused Pol II is not detected. Furthermore, incubation of Lola-I protein with nucleosome-bound sequences in vitro suggested that Lola-I preferentially binds to its motifs near the nucleosome edge when the DNA is partially unwrapped. While this in vitro assay is based on a strong nucleosome-positioning sequence and may not necessarily reflect Lola-I's binding in vivo, Lola-I's binding preference in this assay is consistent with those of other pioneer factors[59,62,69].

We conclude that Lola-I functions similarly to constitutive promoter pioneer factors. Constitutively expressed GAF also preferentially binds to promoters (like Lola-I ~ 60%) and promotes Pol II pausing. In contrast, pioneer factors that regulate enhancers, such as Zelda or Opa, do not bind promoters to the same extent as Lola-I (Zelda ~40% and Opa 35%)[6,70,71], and no effect on paused Pol II prior to transcription has been reported. Notably, the establishment of poised Pol II is largely unaffected in Zelda mutants[70]. Thus, Lola-I has the function of a promoter pioneer factor but its expression is developmentally regulated and thus it changes the state of Pol II pausing at promoters over developmental time.

Since this type of promoter regulation during development has not been observed before, it is interesting to speculate why the Lola-I target genes, which affect metabolism and ion transport, are regulated in this way. One possibility is that Lola-I serves as a timer that makes sure that the target genes are not expressed precociously, while still allowing tissue-specific enhancers to control their expression patterns. Another possibility is that having these promoters open in the early embryo would hijack early patterning enhancers and hinder the activation of patterning genes. This would be consistent with the recent observation that enhancer-promoter contacts change in the late embryo[72]. Moreover, certain BTB domain containing proteins such as GAF have been shown to affect 3D chromatin organization[73]. If so, this could pinpoint to a role of Lola-I and paused Pol II in 3D chromatin organization.

While Lola-I's role as promoter pioneer factor appears unusual, it is possible that other pioneer factors affect promoters in a developmentally regulated way. For example, only a subset of the opening set promoters are regulated by Lola-I. The *lola* locus has many isoforms that are expressed in various patterns during development[32]. They all have the same BTB domain for protein-protein interactions but have different sequence-specific DNA-binding domains. Motifs for some of the isoforms have recently been suggested to regulate chromatin accessibility in the *Drosophila* brain[74]. Thus, it is conceivable that other BTB-domain-containing transcription factors also target promoters and help establish paused Pol II in specific tissues or stages of development.

This likely also holds true for mammalian development. We likely discovered promoter regulation in *Drosophila* because the gene structure is simpler, promoters have high levels of Pol II pausing, and time course experiments can be easily performed. However, the strength of promoters in mammals is also influenced by transcription factor motifs[75], which could modulate the level of accessibility. Furthermore, an opening of promoters has been observed during early mouse development[76]. An even more intriguing possibility is that pioneer factors regulate the usage of alternative start sites. At least 15% of protein-coding genes in the human genome use alternative tissue-specific promoters that are enriched for specific transcription factor binding motifs[77], but the mechanism underlying this promoter selection is not known[30]. It is tempting to speculate that alternative promoter selection during mammalian development is regulated by pioneer factors with analogous roles to Lola-I.

Whether or not the developmental regulation of promoters is widespread, our finding that promoters can be regulated at the level of accessibility further blurs the distinction between promoters and enhancers. The traditional separation between promoters and enhancers has already been challenged in recent years since their function is often not cleanly separable in reporter assays[5,30]. However, the constitutive accessibility of promoters and the more dynamic accessibility of enhancers have still been considered to be distinctive features of this class of elements[5]. The fact that promoters can also be dynamically regulated during development further supports the idea that enhancers and promoters share fundamentally similar characteristics with each other.

## Methods

### Fly stocks and genetic crosses

*Oregon-R* was used as the wild-type strain. Lola-I mutant lines were obtained from Bloomington stock center (ORC4 - 28267) and from Edward Giniger (ORE50). Homozygous *lola-I* mutant files were nonviable and were maintained over a *CyO-GFP* balancer to allow sorting of the homozygous mutant embryos that are GFP⁻. Lola-I rescue lines were generated as follows: a construct with an *Actin* promoter driving full-length *lola-I* cDNA and marked by *mini-white* was integrated into the *attP40* locus on 2 L and then crossed with the *lola-I^ORC4* line (*lola* is on 2 R) to obtain females that recombine the second chromosome in the germ line. After crossing in a *CyO* balancer, several males harboring the *mini-white* marker were selected. After mating single males with a *CyO* balancer stock, each male was screened for the *lola-I^ORC4* mutation by amplifying the relevant portion of the *lola* locus by PCR and sequencing. Meiotic recombinants that had both the rescue construct and the *lola-I^ORC4* mutation were viable as homozygotes. For the INTACT experiments, embryos from fly stocks expressing tissue-specific *RAN-GAP-mcherry-FLAG-BirA* with the help of tissue-specific Gal4 driver lines were collected as described[18]. To isolate tracheal or gut cells from *lola⁻/⁻* embryos, fly lines containing *RAN-GAP-mcherry-FLAG-BirA* on the second chromosome (expressing in either trachea or gut) was recombined with the *lola-I^ORC4* chromosome and maintained over a GFP-marked *CyO* balancer (Trachea: w[*]; P{w[+mC]=GAL4-btl.S}2, P{w[+m*]=lacZ-un8}276,    p[UAS-3xFLAG-blrp-mCherry-RanGap,    UAS-BirA)5,

lola^ORC4^/CyO-GFP) (Gut: w[*]; P{GawB}NP3084, p[UAS-3xFLAG-blrp-mCherry-RanGap, UAS- BirA)5, lola^ORC4^/CyO-GFP). Homozygous embryos for the recombinant chromosome were obtained by sorting for GFP-negative embryos.

## Embryo collection and immunostainings

Adult fly maintenance and embryo collections were performed as described[18]. Briefly, embryos were collected and matured at 25 °C, then dechorionated for 1 min with 67% bleach and cross-linked for 15 min with 1.8% formaldehyde (final concentration in water phase). For the single-molecule FISH experiments, the embryos were cross-linked in 1x PBS in DEPC-treated water. Homozygous lola-I mutant embryos were obtained by sorting for GFP-negative embryos in PBT (PBS with 0.05-0.1% Triton). Embryos used for ChIP-seq experiments were flash frozen in liquid nitrogen and stored at −80 °C. For ATAC-seq and scRNA-seq experiments, the embryos were processed immediately after dechorionation without crosslinking. For immunostainings, antibodies were used in the following dilutions: Lola-I (custom-made by Genescript) 1:750, α-tubulin antibodies (Sigma, T9026) at 1:500, Lamin (ADL40 from Developmental Studies Hybridoma Bank, DSHB, at 1:750), α-MHC (1:500), Elav (7E8A10 from DSHB) at 1:30, and Repo (8D12 from DSHB) at 1:10. For Western blot experiments, antibodies against Lola-I (custom-made by Genescript) were used 1:2000, those against Pol II (Rpb3, custom made from GeneScript, Zeitlinger lab 163185-50) at 1:2000.

## Isolation of tissue-specific nuclei

Nuclei isolation was performed using modified versions of previously published protocols[37,38], as described in ref. 18.

## ChIP-seq experiments

Antibodies were raised against the Lola-I-specific portion (455-877 AA), thus excluding the BTB domain. It also excludes the DNA-binding domains of Lola-I. The antigen for antibody production was expressed in E.coli with a his-tag and purified using a Ni column. The antibody production was done by inoculating the antigens in rabbits and the antibody serum was antigen-affinity purified. Rabbit polyclonal antibodies against the full-length Drosophila Pol II subunit Rpb3 (custom made from GeneScript, Zeitlinger lab 163185-50) is also used.

ChIP-seq experiments were performed as follows. ~100 mg embryos were used per ChIP, and 5 μg chromatin was used for tissue-specific ChIP-seq experiments. Fixed embryos were homogenized by douncing in an ice cold A1 buffer (15 mM HEPES (pH 7.5), 15 mM NaCl, 60 mM KCl, 4 mM MgCl, 0.5% Triton X-100, 0.5 mM DTT, protease inhibitors) and A2 buffer (15 mM HEPES (pH 7.5), 140 mM NaCl, 1 mM EDTA, 0.5 mM EGTA, 1% Triton X-100, 0.1% sodium deoxycholate, 0.1% SDS, 0.5% N-lauroylsarcosine, protease inhibitors) in a tissue grinder for 10–15 times in A1 and A2 buffer each. Then the sonication of the chromatin was performed with a Bioruptor Pico for four-five rounds of 30 s on and 30 s off cycles. The sonicated chromatin was cleared by centrifugation and the supernatant was used for ChIP. Chromatin was incubated with antibodies pre-bound to Dynal magnetic beads (IgA or IgG) overnight with end-to-end rotation at 4 °C and washed with an ice-cold RIPA buffer (50 mM HEPES (pH 7.5), 1 mM EDTA, 0.7% sodium deoxycholate, 1% NP-40 (IGEPAL CA-630), 0.5 M LiCl). Eluted, reverse cross-linked DNA was then purified using phenol-chloroform-isoamylalcohol phase separation and ethanol precipitation. ChIP-seq libraries were prepared from 5 to 15 ng ChIP DNA or 100 ng input DNA according to the manufacturer's instructions (NEBNext ChIP-Seq Library Prep kit).

## ATAC-seq and MNase-seq experiments

ATAC-seq was performed using ~500–2000 embryos of stage 14–17 h AED. Nuclei were isolated by douncing the embryos in the HBS buffer as described above in the Isolation of tissue-specific nuclei section. Whole embryo ATAC-seq was performed without the selection of

nuclei from a specific tissue using the OregonR embryos. The transposition of the nuclei was performed as described in[78]. Computational filtering for fragments of size 0–100 bp was done to capture signals from the accessible regions.

MNase digestion was performed similarly to previously published protocols[79]. Briefly, chromatin was extracted from 0.1 mg of Oregon-R or lola-I mutant embryos per replicate by douncing embryos in the NPS buffer (0.5 mM spermidine, 0.075% IGEPAL CA-630, 50 mM NaCl, 10 mM Tris-Cl (pH 7.5), 5 mM MgCl, 1 mM Cacl, 1 mM beta-mercaptoethanol) using a tissue homogenizer, then digested with a concentration gradient of MNase (Worthington Biochemical Corporation #LS004798) in NPS buffer for 30 min at 37 °C. All digestion concentrations were run on a gel and the concentration to be sequenced was chosen such that the digestion is complete, characterized by the presence of only mononucleosomes, but the samples are not over digested (no smaller than mononucleosome sized fragments). Libraries were prepared from purified MNAse digested DNA using the NEBNext DNA Library Prep kit following the manufacturer's instructions and then paired-end sequenced on an Illumina HiSeq 2500 sequencing system. Computational filtering for fragments of size 100–200 bp to analyze the nucleosome occupancy.

## Nucleosome binding assay

Full-length Lola-I protein was expressed using baculovirus and purified by Genescript. Briefly, the lola-I sequence was synthesized and subcloned into the Flag-TAG expression vector F1 and expressed in Sf9 insect cells using a recombinant Bacmid. Sf9 cells were grown in Sf-900 II SFM Expression Medium in Erlenmeyer Flasks at 27 °C in an orbital shaker. Cells were seeded in 6 wells, transfected the next day by adding DNA mixed with Cellfectin II at an optimal ratio, and incubated for 5–7 days before harvesting P1 and P2 viral stock. The Sf9 cells (1 L) containing 5% FBS were infected with the P2 virus at MOI = 3 and harvested at 48 h post-infection. Cells were sonicated in 50 mM Tris, pH 8.0, 150 mM NaCl, 5% Glycerol containing protease inhibitors. Cell pellets were harvested and lysed, and the supernatant was incubated with Flag Columns to capture the target protein. Fractions were pooled and dialyzed with 50 mM Tris-HCl, 150 mM NaCl, 5% Glycerol, pH 8.0 followed by 0.22 um filter sterilization. Proteins were analyzed by SDS-PAGE and Western blot by using standard protocols for molecular weight and purity measurements.

The nucleosome binding assay was performed as previously described[62]. Templates were designed by altering the right side of the Widom 601 nucleosome-positioning sequence and placing Lola-I binding motifs with increasing distance to the dyad axis. Four translational settings were tested—dyad (at superhelix location (SHL) R0, R0.5), intermediate (SHL R4, R4.5), edge (SHL R6, R6.5, R7), and linker, which is outside the nucleosome (SHL 8). The rotational setting of each motif was designed such that it is either outside on the solvent accessible side (SHL R0.5, R4.5, R6.5) or concealed (SHL R0, R4, R6, R7) based on the nucleosome crystal structure formed on the Widom 601 sequence[80]. To explore cooperativity, a template with two neighboring motifs was designed (SHL R6 + R7), as well as one with two motifs further apart as a control (SHL L6 on left +R7). All templates were compared to non-specific binding to the Widom 601 sequence.

All 11 synthesized DNA templates were amplified via PCR with the primer pair 5′-GATGGACCCTATACGCGGC-3′ and 5′-GGAACACTATCC GACTGGCA-3′, and the products were column-purified (QIAGEN), quantified, and pooled equally. In vitro nucleosomes were generated from H2A/H2B dimer and H3.1/H4 tetramer (NEB). The pool of 11 nucleosome sequences were added to histones at octamer/DNA molar ratios of 1.25:1 in 2 M NaCl solution. Nucleosomes were reconstituted by salt gradient dialysis as described in[81], purified from free DNA with 7–20% sucrose gradient centrifuge, and concentrated by 50 K centrifugal filter units (Millipore, AmiconR Ultra).

For the protein-nucleosome binding assays, each of 0.25 pmol of purified nucleosomes were incubated with increasing concentrations of Lola-I protein (molar ratios of 0:1, 1:1, 2:1, 4:1 to 10:1) in 7 μl DNA-binding buffer (10 mM Tris-Cl, pH7.5; 50 mM NaCl; 1 mM DTT; 0.25 mg/ml BSA; 2 mM MgCl2; 0.025% P-40; and 5% glycerol) for 10 min on ice and then for 30 min at room temperature. Protein binding was detected by mobility shift assay on 4% (w/v) native polyacrylamide gels (acrylamide/bisacrylamide, 29:1, w/w, 7 × 10 cm) in 0.5 × Tris Borate-EDTA buffers at 100 V at 4 °C. After electrophoresis, DNA was imaged by staining with SYBR Green (LONZA).

All visual bands were excised from the gel, as well as the bands at the same locations in the other lanes. Each gel slice was processed separately for a total of 20 samples from 2 replicate experiments. In order to extract DNA from polyacrylamide gel, the chopped gel slices were soaked in diffusion buffer (0.5 M ammonium acetate; 10 mM magnesium acetate; 1 mM EDTA, pH8.0; 0.1% SDS), and incubated at 50 °C overnight. The supernatant was collected, residual poly-acrylamide removed with glass wool, and DNA purified with QIAquick Spin column (QIAGEN). The DNA concentration for each sample was determined by qPCR by comparing it to a standard curve generated from the control 601 sequence. Based on this concentration, the samples were amplified by PCR using Illumina primers (the cycle number ranged from 8 to 12) and then indexed in a second round of PCR using Nextera dual index primers, followed by clean up with AMPure XP beads (Beckman Coulter). The samples were multiplexed and sequenced on an Illumina MISeq using 2 × 150 bp paired-end sequencing. Sequencing and quality control were performed at the University at Buffalo Genomics and Bioinformatics Core.

High-quality sequence reads were mapped to each specific starting sequence using VSearch[82]. After obtaining the amount of the reads from each band, the data from each of the nucleosome sequences N were normalized to the Widom 601 control sequences: The results were then analyzed relative to the 601 control sequence and the non-specific binding in the input lane without any Lola-I. Relative shift is determined from the non-shifted nucleosome bands and controls for the technical variability introduced by gel-excision, PCR, NGS-library construction, or NGS sequencing. In this method each specific nucleosome sequence or the super-shifted sequence is measured relative to non-specific binding (601 fragment without a Lola-I binding site):

$$\text{Relative shift} = -\log_2((\text{reads nuclesome}_N/\text{reads nucleosome}_{601})/$$
$$(\text{reads nucleosome input}_N/\text{reads nucleosome input}_{601}))$$

where N is one of the 11 nucleosome sequences, 601 is the control nucleosome sequence, reads nucleosome is the nucleosome band at a specific concentration of Lola-I, reads nucleosome input is the nucleosome band in the input lane without any Lola-I added. Or as,

$$\text{Relative shift} = -\log_2((\text{reads supershift}_N/\text{reads supershift}_{601})/$$
$$(\text{reads supershift input}_N/\text{reads supershift input}_{601}))$$

where N is one of the 11 super-shifted sequences, 601 is the control nucleosome sequence, reads supershift is the super-shifted band at a specific concentration of Lola-I, reads supershift input is the super-shifted band in the input lane without any Lola-I added.

## Single-molecule FISH experiments

Stellaris single-molecule FISH probes were designed for the *Drosophila melanogaster* genes *Gip*, *PPO1*, *PPO2*, and *lola-I* using the Stellaris probe designer, and purchased with a label ready C-term TEG-Amino tag from Biosearch Technologies. 4 nMol of *PPO1* and *PPO2* probe sets were combined and labeled with two units of AF647 amine reactive succinimidyl ester Decapacks (ThermoFisher) and the *Gip* probe set

was labeled similarly, with AF555 and *lola-I* with AF647. Labeling was overnight at 4 °C in 0.1 M sodium tetraborate at pH 9. HPLC was used to purify labeled from unlabelled probes as described[83].

The single-molecule FISH technique was optimized and adapted from the Stellaris website and based on Dr. Shawn Little's protocol (personal communication). Embryos were collected, dechorionated with bleach and crosslinked in 4% formaldehyde in a 1:3 mixture of PBS and heptane. Embryos were then sorted in PBT (0.1% Triton), and then devitellinated with a 1:1 mixture of methanol and heptane, equilibrated in methanol and stored at −20 °C. After gradual rehydration, embryos were post-fixed, treated with proteinase K (0.5 ug/ml) for 1 h on ice and 1 h at 37 °C, and crosslinked once more. The permeabilized embryos were washed with a series of PBT and Stellaris WashA buffers. The embryos then underwent prehybridization in the Stellaris Hyb buffer for one day at 37 °C. Hybridization with 1 μM concentration of probes was performed at 25 °C overnight. After several washes with WashA buffer and one with PBS, DAPI staining was performed in PBS buffer (5 μg/ml) for 5 min, followed by washes with PBT buffer to remove unbound DAPI. Embryos were mounted in Prolong gold and cured at room temperature (22 °C) for 3 days to a week.

For analyzing *lola-I* transcripts, images of whole-mount *Drosophila* embryos were acquired on a PerkinElmer Ultraview spinning disk confocal microscope equipped with an EM-CCD camera (model C9100-13; Hamamatsu Photonics), using Volocity software (PerkinElmer). An Apochromat 63×, 1.46 NA oil immersion objective was used with a 405/488/561/640 nm multiband dichroic. Dual color images of DAPI and *lola-I*-Af647 were acquired with 405 nm and 640 nm laser lines, respectively, with 415–475 nm emission filter for DAPI and a 660–750 nm emission filter for Af647.

For the expression analysis, images of whole-mount *Drosophila* embryos were acquired from a Nikon 3PO spinning disc, with a W1 disc, sitting on a Nikon Ti Eclipse base, controlled by Nikon Elements. Data were collected either with a 40×1.1 NA Plan ApoChromatic long working distance water objective (for overviews), or a 100X, 1.4 NA Plan-Apochromatic oil objective (for observation of nascent transcripts, single transcripts, and for transcriptional modeling). AF647, AF555, and DAPI were excited with 640 nm, 561 nm, and 405 nm lasers, respectively, through a 405/488/561/640 nm main dichroic. Emission filters included a 700/75- and 455/50 nm dual band pass filter for the far red and DAPI channel, and a 605/70 nm emission filter for the red channel. An ORCA-Flash 4.0 V2 digital sCMOS camera was used, Z-step spacing was 1 micron for the 40× overview data, and 0.3 microns for 100× data. For 40× data, prior to display, gut autofluorescence was subtracted from the red channel using a reference in the green channel.

For quantification of transcripts per cell for transcriptional modeling, the 100× data were used. A Gaussian blur of radius 1 pixel was applied, followed by a rolling ball background subtraction with a radius of 50 pixels. To integrate the total signal over each cell, a z bin of 7 was applied for a final spacing of 2.1 microns. Cell outlines were manually drawn in FIJI, and integrated intensity was taken over 3 of the binned z slices, for a total cell size in z of 6.3 microns, after an intensity threshold was applied to remove the background. This background intensity was found from areas with no visible transcripts. Total cells counted were 184 from 7 wild-type embryos, and 340 cells from 14 *lola^{ORC4}* embryos of 12–14 h. To calibrate transcripts per cell from integrated intensity, single transcripts were fit to a two-dimensional Gaussian. The total integrated intensity per cell was then divided by the average integrated intensity of single transcripts to get the number of transcripts per cell. Likewise, to find the intensity of nascent sites, nascents were identified and fit to a two-dimensional Gaussian. The integrated intensity of the Gaussian was then divided by a single RNA spot intensity to yield the number of RNAs per nascent.

The fit of the distributions of transcripts per cell to the simple two-state model shown in Supplementary Fig. 3c was done as described[52]:

$$p(N) = (\Gamma(K_{on}+N) * \Gamma(K_{on}+K_{off}) * K_{prod}^{\wedge}N)/(\Gamma(N+1) * \Gamma(K_{on}+K_{off}+N) * \Gamma(K_{on}))$$
$$HG(K_{on}+N, K_{on}+K_{off}+N, -K_{prod})$$

$$(1)$$

where $HG$ is the confluent hypergeometric function of the first kind, $K_{on}$ is the activation rate constant, $K_{off}$ is the inactivation rate constant, and $K_{prod}$ is the production rate constant, all expressed as ratios to the degradation rate constant. Given the difficulty in fitting these complex distributions, we took advantage of the fact that we can observe nascent transcripts for transcriptionally active cells. That lets us measure the ratio of active to inactive cells and therefore the ratio of activation to inactivation rates:

$$\frac{(Nascent\ Cells)}{(Empty\ Cells)} = \frac{Active}{Inactive} = \frac{Kon}{Koff}$$

$$(2)$$

where this ratio is fixed in the analysis to calculate $K_{off}$ from $K_{on}$. Fitting was accomplished with the scipy optimize curve_fit function in python utilizing the trf (trust region reflective) algorithm[84,85]. Because the variance is higher for more frequent histogram bins and those are the bins that are most confidently determined for our data set, fitting was accomplished without weighting. Errors are determined by Monte Carlo analysis. This analysis assumes that the data distribution is reasonably well described by the fit and simulates random distributions repeatedly from the fit distribution and fits those distributions to obtain the error distribution for the fit parameters. For wild-type data, the fit produced the following parameters relative to the degradation rate constant: an activation rate constant, $K_{on}$, of $2.7 \pm 0.42$ and a production rate constant, $K_{prod}$, of $1524 \pm 58$. From the images, we know the ratio of active/inactive is 0.69. Taking these data together yields an inactivation rate constant, Koff, of 3.7. For the mutant, the fit produced a $K_{on}$ of $0.19 \pm 0.12$ and a $K_{prod}$ of $571 \pm 207.8$. From the images we know that the ratio of active/inactive is 0.036. These values yield a $K_{off}$ of 5.1. Supplementary Data 4 shows the values of these parameters and expected values based on reasonable literature mRNA half-lives.

## mRNA-seq and scRNA-seq experiments
mRNA-seq and scRNA-seq experiments were performed as described[18,86] with isolated RNA from entire embryo or single cells from 14 to 14.5 h AED *Oregon-R* embryos or *lola-I* mutant embryos sorted for homozygous mutant embryos that are GFP⁻. Total mRNA was extracted from non-cross-linked embryos using the Maxwell Total mRNA purification kit (Promega, #AS1225). mRNA from single cells for the scRNA-seq experiments was isolated using the 10x Genomics instrument.

## Antibodies used
Lola-I (custom-made by Genescript), α-tubulin antibodies (Sigma, T5168), Lamin (ADL40 from Developmental Studies Hybridoma Bank, DSHB, at 1:750), α-MHC (source not tractable but validated by immuno-staining experiments; also these are only used in the supplements to support a minor point), Elav (7E8A10 from DSHB), Repo (8D12 from DSHB), Pol II (Rpb3, custom made from GeneScript, Zeitlinger lab 163185-50).

## Sequence alignment
Reads from the ChIP-seq data and ATAC-seq experiments were aligned to the *Drosophila* melanogaster genome (dm6) using Bowtie (v 1.1.2)[87], allowing a maximum of two mismatches and including only uniquely aligned reads. Coverage files were created by extending the aligned reads to the estimated insert size or the actual size for the paired-end

libraries. For the bulk mRNAseq samples, pseudo-alignment was performed using the Kallisto package (0.46.0)[88], to calculate the gene expression values. For the scRNA-seq samples, alignment and separations of reads from different cells and quantification of gene expression were done using the Cell Ranger pipeline (v 2.1.1) from 10× Genomics.

## Analysis of single-cell RNA-seq data
The wild-type scRNAseq data were aligned to the *lola*⁻/⁻ scRNAseq data by performing the canonical correlation analysis (CCA) using the Seurat package[89,90]. Gene expression in a particular tissue/ single-cell cluster is measured either using the median absolute normalized expression counts or using the percentage of cells with any detectable transcripts for each gene in each tissue. Expression of each Lola-I target gene was calculated in the presumed target tissue (sc-cluster with the maximum expression for each gene was considered as the expressing tissue for that gene) and the presumed non-expressing tissues (the five least expressing sc-clusters for each gene were considered as other tissues for that gene).

## Gene groups
Control genes are a randomly chosen 250 gene-subset of the constant set genes (Gaertner et al., 2012) (Supplementary Data 2). Lola-I binding sites are divided into promoter proximal ($n = 330$) and distal sites ($n = 233$) based on whether the binding sites are present within +50 to −450 bp of an annotated TSS (flybase version r6.21) (Supplementary Data 2). For comparisons with other pioneer factors (GAF and Opa[71]) we used the same promoter and distal site definitions for our analysis or directly used the frequencies from the publication when the definitions are comparable (Zelda[6]). Lola-I targets are defined as genes with a Lola-I binding peak (using the MACS2 peak caller) in both the Lola-I ChIP-seq replicates in the promoter region (Supplementary Data 2) ($n = 329$). Lola-I targets genes are further filtered for at least a twofold change in pol II occupancy between wild-type and mutant embryos for the scRNAseq analysis.

## Motif enrichment analysis
De-novo identification of the Lola-I motif (position weight matrix) found at the opening-set genes in Fig. 1b was done using the MEME package (MEME 5.0.5) (meme macs_peaks_late_genes.fasta -oc macs_peaks_late_genes_meme_output -p 5 -mod zoops -dna -nmotifs 10 -revcomp -maxw 12 -maxsize 5000000). Enrichment analysis of the Lola-I motif was done by scanning for the AAAGCTY motif. Unbiased enrichment analysis of all known motifs from the MotifDb package were scanned using the FIMO package and then the enrichments for each of the motifs was tested using a chi-squared test.

## Statistical significance calculations and data visualization
*P* values in Figs. 1d, g, 3a, b, 5a, b, c and Supplementary Figs. 1b, 2b, 2c, 3c, 6a were calculated with the two-sided Wilcoxon test. *P* values in Fig. 1a were calculated with the chi-squared test with multiple-testing correction. *P* values in Supplementary Data 3 were calculated using the hypergeometric test. The replicate correlation values measured by Pearson correlation coefficient is given in Supplementary Data 5. Heat maps were normalized, and really low or high values were ceiled or floored, respectively. In the heat map visualizations, ChIP-seq enrichments within a region were normalized to the 80th percentile (set as maximum) for a balanced view of all values, and ChIP-seq enrichments less than twofold over the input are set to zero (minimum). MNase-seq and ATAC-seq signals are normalized to the 85th percentile, which gives a sufficiently balanced view of the data. The rows in heatmaps were ordered based on the associated flybase id alphabetically, unless mentioned otherwise. Box plots show the median as the central line, the first and the third quartiles as the box, and the upper and lower whiskers extend from the quartile box to the largest/smallest value

within 1.5 times of the interquartile range. The outliers were not shown in box plots.

### Reporting summary

Further information on research design is available in the Nature Portfolio Reporting Summary linked to this article.

## Data availability

Raw and processed functional genomics data associated with this manuscript have been deposited in GEO under accession number, "GSE200875". Part of the wild-type functional genomics data is available from the previously released GEO dataset, "GSE120157". Source data for the graphs are provided with this paper. The nucleosome model figure uses the nucleosome structure from the RCSB PDB database, "5NL0". The raw and processed data are also available through a publicly accessible Amazon Linux virtual machine image (ami-id: ami-013fa11a4a9b52628) (https://aws.amazon.com/console/) and the unprocessed microscopy and western blot images are available through the stowers institute's original data repository (https://www.stowers.org/stowers-odr). Source data are provided with this paper.

## Code availability

All data analysis performed in this paper, including raw data, processed data, software tools, and analysis scripts are available through a publicly accessible Amazon Linux virtual machine image (ami-id: ami-08731b3f99f24143f)(https://aws.amazon.com/console/). The analysis code is also available on GitHub at https://github.com/zeitlingerlab/Ramalingam_Lola_2022.git and on Zonedo (https://doi.org/10.5281/zenodo.8112025)[91].

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

## Acknowledgements

We thank Cindi Staber for the help in designing and coordinating the Lola-I protein and Lola-I antibody production with Genescript. We thank Shawn Little for sharing his single-molecule FISH protocol. We thank Kate Hall, Allison Peak, and Ana Pinson for the technical assistance with scRNA-seq experiments. We thank Divya Krishna Kumar for the technical assistance with single-molecule FISH experiments. We thank Robb Krumlauf and Mounia Lagha for their feedback on the paper. This work was supported by funding from the National Institutes of Health (DP2OD004561) and the Stowers Institute for Medical Research. This work was done to fulfill, in part, requirements for Vivekanandan Ramalingam's PhD thesis research as a student of the University of Kansas.

## Author contributions

V.R. and J.Z. conceived the project, designed the study and wrote the paper. V.R. performed all the experiments and the analysis under the supervision of J.Z. except for the following: X.Y. under the supervision of M.B., performed the nucleosome binding assays, including the library preparation and the analysis of the data. B.S., J.U. and J.J.L. helped with the single-molecule FISH experiments, the image acquisition and performed the analysis of the data, K.J.B. analyzed the *lola-I* mutant embryos using immunostainings and Western blots, A.O. helped to develop the single-molecule FISH protocol, M.N. helped with the tissue-specific ChIP-seq experiments. All authors provided input on the paper.

## Competing interests

The authors declare no competing interests.
