## [Peer Review File · Nature Communications]

Lola-I is a promoter pioneer factor that establishes *de novo* Pol II pausing during developmentEditorial Note: This manuscript has been previously reviewed at another journal that is not operating a transparent peer review scheme. This document only contains reviewer comments and rebuttal letters for versions considered at Nature Communications.

REVIEWERS' COMMENTS

Reviewer #1 (Remarks to the Author):

The authors have done a nice job responding to my questions and suggestions, and I think the revisions to emphasize the significance of the work are indeed useful.

Reviewer #2 (Remarks to the Author):

The authors have thoroughly addressed all prior critiques, resulting in a well-written and compelling manuscript that identified Lola-I as a pioneer factor essential for establishing paused RNA Polymerase II at the promoters of developmentally regulated genes.

A few minor issues were noted:

1. On line 55, the authors write “We identified the responsible transcription factor . . .”. I would suggest amending this to “We identified a responsible transcription factor . . .” since Lola-I does not appear to be the only responsible factor by which promoters acquire Pol II de novo.
2. Throughout the manuscript there are multiple examples where *lola* is not italicized, but should be when referring to a locus, gene, or mRNA. For example, lines 69, 86,88,89,92, and many more.
3. It is a bit confusing that the Supplementary Figures are referenced out of order.
4. While addressed in the response to reviewers, it would be helpful to note how heatmaps are ordered in the figure legends so it is immediately clear.
5. In Figure 3A, it is not noted what blue is indicating in the image.
6. The citation to Blythe and Wieschaus 2015 on lines 273-274 appear incorrect. I imagine the citation being considered is regarding Opa and is Soluri et al. eLife 2020 from the Blythe lab.
7. In the citations, Sun et al. is inadvertently cited twice.

Response to reviewer:

Reviewer #1 (Remarks to the Author):

The authors have done a nice job responding to my questions and suggestions, and I think the revisions to emphasize the significance of the work are indeed useful.

Thank you.

Reviewer #2 (Remarks to the Author):

The authors have thoroughly addressed all prior critiques, resulting in a well-written and compelling manuscript that identified Lola-I as a pioneer factor essential for establishing paused RNA Polymerase II at the promoters of developmentally regulated genes.

A few minor issues were noted:

1. On line 55, the authors write “We identified the responsible transcription factor . . .”. I would suggest amending this to “We identified a responsible transcription factor . . .” since Lola-I does not appear to be the only responsible factor by which promoters acquire Pol II de novo.

It is now modified as “Here, we identify one of the responsible transcription factors as Lola-I . . .”.

2. Throughout the manuscript there are multiple examples where lola is not italicized, but should be when referring to a locus, gene, or mRNA. For example, lines 69, 86,88,89,92, and many more.

Done

3. It is a bit confusing that the Supplementary Figures are referenced out of order.

We understand and agree with the concern, but we did put in a lot of effort to arrange the supplementary figures in both in the order of their appearance in the text and also group them into a logical theme. However, it is not always possible to arrange them to match the order in the main text.

4. While addressed in the response to reviewers, it would be helpful to note how heatmaps are ordered in the figure legends so it is immediately clear.

We appreciate the suggestion, however, due to space limitations in the legends we cannot include more text in the legends.

5. In Figure 3A, it is not noted what blue is indicating in the image.

Done

6. The citation to Blythe and Wieschaus 2015 on lines 273-274 appear incorrect. I imagine the citation being considered is regarding Opa and is Soluri et al. eLife 2020 from the Blythe lab.

Done

7. In the citations, Sun et al. is inadvertently cited twice.

Done